# Pre-Existing Hypertension Is Related with Disproportions in T-Lymphocytes in Older Age

**DOI:** 10.3390/jcm11020291

**Published:** 2022-01-06

**Authors:** Anna Tylutka, Barbara Morawin, Artur Gramacki, Agnieszka Zembron-Lacny

**Affiliations:** 1Department of Applied and Clinical Physiology, Collegium Medicum University of Zielona Gora, 65-417 Zielona Gora, Poland; a.tylutka@cm.uz.zgora.pl (A.T.); b.morawin@cm.uz.zgora.pl (B.M.); 2Faculty of Computer, Electrical and Control Engineering, Institute of Control and Computation Engineering, University of Zielona Gora, 65-417 Zielona Gora, Poland; a.gramacki@iss.uz.zgora.pl

**Keywords:** hypertension, immune risk profile, elderly, prevention of cardiovascular disease

## Abstract

Age-related immune deficiencies increase the risk of comorbidities and mortality. This study evaluated immunosenescence patterns by flow cytometry of naïve and memory T cell subpopulations and the immune risk profile (IRP), expressed as the CD4/CD8 ratio and IgG CMV related to comorbidities. The disproportions in naïve and memory T cells, as well as in the CD4/CD8 ratio, were analysed in 99 elderly individuals (71.9 ± 5.8 years) diagnosed with hypertension (*n* = 51) or without hypertension (*n* = 48), using an eight-parameter flow cytometer. The percentage of CD4^+^ T lymphocytes was significantly higher in hypertensive than other individuals independently from CMV infections, with approximately 34% having CD4/CD8 > 2.5, and only 4% of the elderly with hypertension having CD4/CD8 < 1. The elderly with a normal BMI demonstrated the CD4/CD8 ratio ≥ 1 or ≤ 2.5, while overweight and obese participants showed a tendency to an inverted CD4/CD8 ratio. CD4/CD8 ratio increased gradually with age and reached the highest values in participants aged >75 years. The decline in CD4^+^ naïve T lymphocytes was more prominent in IgG CMV+ men when compared to IgG CMV+ women. The changes in naïve and memory T lymphocyte population, CD4/CD8, and CMV seropositivity included in IRP are important markers of health status in the elderly that are dependent on hypertension.

## 1. Introduction

Age-related changes in the immune system have been studied extensively over the past years, however, differences between immune cells in the young and the elderly are not yet entirely clear [1]. Loss of lymphoid tissue and impairment of the immune system are observed with age. These changes are commonly referred to as immunosenescence, and are associated with the increased susceptibility to a number of diseases, including cardiovascular and autoimmune diseases, as well as an impaired response to vaccination [2]. One of the most important age-related changes in the adaptive immune system is the involution of the thymus, the sole organ responsible for the production of T lymphocytes [3]. During the aging process, the population of T lymphocytes expressed as CD45RA^+^ decreases while the population of CD45RO^+^ T lymphocytes intensively proliferates, thereby reversing the lymphocyte balance in naïve and memory T lymphocytes [4,5]. Different mechanisms of innate and acquired immunity in genders correspond to different responses to both new and self-antigens in men and women. Immunologically, immunosenescence and inflammaging appear to be more profound in older males when compared to females [6]. Androgens polarize naïve CD4^+^ T lymphocytes towards the Th1 subset, contrary to estrogens that stimulate the Th2 response and activate the production of antibodies [7]. A decrease in the levels of androgens and estrogens contributes to thymic involution, reducing the number of naïve T lymphocytes [8]. Many studies related to immunosenescence have been conducted in the last decade, and three prospective studies have assessed the immune risk profile (IRP) in the elderly, defined by the CD4/CD8 ratio. The CD4/CD8 ratio was found to increase with age in OCTO/NONA surviving participants over 100 years of age [9]. On the other hand, the analysis by Vasson et al. [10] showed a decreasing trend of the CD4/CD8 with age in Spanish and French populations. Guzik et al. [11] discovered for the first time that T-lymphocytes are an important determinant of hypertension and were among the first to emphasize that hypertension is an immunological disease. An adaptive response of CD4^+^ T cells has been implicated in hypertension and its complications, including atherosclerosis-based cardiovascular diseases. Moreover, according to Avinas et al. [12], by simply detecting the total numbers of peripheral CD4^+^ T and CD8^+^ T cells and their ratio the development and progression of hypertension and heart failure could be predicted. Therefore, this study was designed to evaluate the effect of common comorbidities in older age, including hypertension, on changes in the T cell subpopulation and to determine the direction of changes in CD4/CD8 ratio and CMV infections in the Polish population in those over 60 years of age.

## 2. Materials and Methods

### 2.1. Participants

The study included one hundred and twenty seniors from the University of the Third Age (U3A), which is a teaching institution for retired individuals whose main goal is to improve the quality of life of the elderly. The organization encourages seniors to participate in various activities and programs such as art classes, computer and graphic arts courses, or dance and theatre. The participants’ current health status was assessed using a health questionnaire [13]. The inclusion criteria were as follows: an informed consent signed by all the participants, 60–90 years of age to be classified as an older group, their mobility, no hospitalisation during the previous 6 months before the study, and the same access to medical care. We based classification on medical records and the medical interview performed by the physician who was engaged in the study, and classified participants into a given group (hypertension and control). On the basis of the medical history, the study excluded elderly individuals with neurological disorders, acute infectious and oncologic diseases, or those with an implanted cardiac pacemaker (body composition analysis could not be performed). Due to hospitalization, serious injuries, or a cold, twenty-three participants withdrew from further participation in the study. Eventually, ninety-nine seniors aged 60–90 years (females *n* = 83, males *n* = 16) were qualified for the study. The older group included 51 persons with hypertension and controls with other diseases, including 32 persons with thyroid diseases only, 9 only with rheumatoid arthritis, and 7 with type 2 diabetes (Figure 1). According to the WHO protocol, the elderly participants were divided into two categories: the youngest-old: aged 60–74 years (*n* = 63) and the middle-old/oldest-old: aged 75–90 years (*n* = 36) [14,15]. The Bioethics Commission at the Regional Medical Chamber Zielona Gora, Poland approved the study (No. 21/103/2018) in accordance with the Helsinki Declaration. Signing an informed consent for participation in the study by each participant was also a prerequisite for their inclusion.

### 2.2. Body Composition

Body composition analysis, including the assessment of fat free mass (FFM) and fat mass (FM), were performed using the bioelectrical impedance method with the Tanita Body Composition MC-980 analyzer (Tanita, Tokyo, Japan) calibrated prior to each test session, in accordance with the manufacturer’s guidelines. Measurements were taken for each individual twice in a standing position between 7:00 and 9:00 am before their first meal. The participants were measured before blood sampling, after a night of rest, and with an empty bladder. The participants were instructed not to exercise vigorously for 12 h prior to the analysis, as the time was needed for the purpose of recovery. The repeatability of the measurements was 98%. We followed the methods of Tylutka et al. [16]. According to the guidelines of Vanallie et al. [17], the ratio of fat free mass index (FFMI) and fat mass was calculated, where FFMI = FFM (kg)/height (m^2^) and FMI = FM (kg)/height (m^2^). The sum of FFMI and FMI gives the body mass indices (BMI). On the basis of the obtained BMI values and following WHO guidelines, a normal body weight was determined where BMI ranged from 18.5 to 24.9 kg/m^2^, overweight meant that the values ranged from 25 to 29.9 kg/m^2^, and obesity the values reached ≥30 kg/m^2^.

### 2.3. Blood Sampling

Fasting blood samples were collected from the median cubital vein in the morning between 8.00 and 10.00 using S-Monovette tubes (Sarsted AG & Co. KG, Nümbrecht, Germany). All the blood samples were placed into specimen tubes containing EDTA-K_2_ (dedicated to morphology and flow cytometry) and were immediately analysed. For the other biochemical analyses, blood samples were allowed to clot (approximately 45 min) and then centrifuged at 3000 rpm for 10 min. Aliquots of serum were stored in a refrigerator at −80 °C.

### 2.4. Flow Cytometry Analysis

The CyLyse kit from Sysmex (Sysmex Europe Gmbh, Norderstedt, Germany) was used for flow cytometry analysis with the eight-parameter CyFlow Space Sorter flow cytometer by SysmexPartec (Sysmex Europe Gmbh, Norderstedt, Germany). A mix of monoclonal antibodies conjugated with fluorochromes (CD8 APC, CD4 FITC, CD45 RA Pacific Blue™, CD45RO PE) was added to 100 µL of the donated blood and incubated in the dark at room temperature for 15 min. Then, 100 µL of Leukocyte Fixation Reagent A was added and incubated again in the dark for 10 min. After the last incubation, 2.5 mL Erythrocyte Lysing Reagent B was added, mixed, and incubated in the dark for 20 min. After this time, the measurements were made. We followed the methods by Tylutka et al. [16]. T helper lymphocytes CD4^+^ and cytotoxic lymphocytes CD8^+^ are expressed as a percent of gated lymphocytes. Memory and naïve subpopulations were gated by positive surface staining for CD45RO and CD45RA, respectively. The strategy of gated lymphocytes T is shown in Figure 2. The ratios of CD4CD45RA/CD4CD45RO and CD8CD45RA/CD8CD45RO were calculated according to Hang et al. [18]. The reference values for CD4/CD8 ratio were adopted from McBride and Striker [19] and Strindhall et al. [9]. The ratios ≥1 or ≤2.5 are generally considered normal, nevertheless there is quite a lot of variability due to past or present infections, age, genetics, ethnicity, and environmental exposure. The CD4/CD8 ratio < 1 and >2.5 is regarded as an immune risk phenotype and can be associated with immunosenescence and chronic inflammatory diseases [9].

### 2.5. Cytomegalovirus (CMV) IgG

The analysis of IgG CMV antibodies was performed by means of the enzyme immunoassay test from DRG International (Springfield Township, Cinninnati, OH, USA). The reference values of CMV-seronegativity (IgG CMV−) were set at <9 DU/mL and CMV− seropositivity (IgG CMV+) at >11 DU/mL. The intra-assay CV for the CMV kit was 7.75% and inter-assay CV was 11.45%. We followed the methods of Tylutka et al. [16].

### 2.6. Haematological Variables

Peripheral blood morphology was assessed using 3 diff BM HEM3 Biomaxima (Lublin, Poland). The following variables were determined: leukocytes, granulocytes (GRA%), lymphocytes (LYM%), red blood cell count (RBC), haemoglobin (HB), haematocrit (HCT), mean corpuscular volume (MCV), mean corpuscular haemoglobin (MCH), mean corpuscular haemoglobin concentration (MCHC), and platelets (PLT). We followed the methods of Tylutka et al. [16].

### 2.7. Biochemical Variables

Using the BM200 Biomaxima analyser (Poland), the determination of total cholesterol (TC), high-density lipoprotein (HDL), low-density lipoprotein (LDL), and triglycerides (TG) was performed. Non-HDL was calculated according to the following formula: non-HDL = TC-HDL. Oxidised low-density lipoprotein (oxLDL) was determined by using ELISA kits from SunRed Biotechnology Company (Shanghai, China) with a detection limit at 3.03 mg/dL. C-reactive protein (CRP) was determined in duplicate with the DRG International (Springfield Township, Cinninnati, OH, USA) highly sensitive enzyme immunoassay test with the detection limit of 0.001 mg/L. The intra-assay coefficient of variation (intra-assay CV) for the CRP ELISA kit was 4.44%, and the inter-assay coefficient of variation (inter-assay CV) was 3.28%. The serum glucose was determined using the DP 310 Vario II mobile spectrophotometer (Berlin, Germany). We followed the methods of Tylutka et al. [16].

### 2.8. Statistical Analysis

All the analyses were performed by using the RStudio, version 1.4.1103 [20]. The normality of the distributions was evaluated using the Shapiro-Wilk test to verify assumptions for the use of parametric and non-parametric tests. The measurements in groups were compared by the one-way ANOVA or the Kruskal-Wallis non-parametric test (if the normality assumption was violated). The analysis of covariance (ANCOVA) was used in classification of body composition and gender that might influence the concentration of T lymphocyte phenotypes in individuals with diseases. Additionally, eta-squared for ANOVA (*η*^2^) was used as a measure of the effect size, which is indicated as having no effect if 0 ≤ *η*^2^ < 0.01, a minimum effect if 0.01 ≤ *η*^2^ < 0.06, a moderate effect if 0.06 ≤ *η*^2^ < 0.14, and a strong effect if *η*^2^ ≥ 0.14 [21,22]. The effect size for the Kruskal-Wallis test as the eta-squared (*η*^2^) was calculated based on the function in Rstudio: Kruskal effsize and = 0.01–0.06 was a small effect, 0.061–0.14 was a moderate effect, and >0.14 was a large effect [23]. Spearman’s rank correlation (r*_s_* Spearman rank correlation coefficient) was used to investigate the relationships between biochemical variables and immune cells. Statistical significance was set at *p* < 0.05. The results are expressed as mean and standard deviation (Mean ± SD) and as a median (Me).

## 3. Results

### 3.1. Body Composition

The body mass index (BMI) ranged from 19.00 to 36.6 kg/m^2^. Approximately 31.3% of the study seniors had normal body mass (18.5–24.9 kg/m^2^), 49.5% were classified as overweight (25–29.9 kg/m^2^), and 19.2% as obese (≥30 kg/m^2^). (Table 1 and Table 2). More than 60% of elderly men and 35% women had FM% higher than the references value. In participants diagnosed with hypertension, more than 50% had higher value of FM%, which can emphasize the role of adipose tissue in the development of hypertension and metabolic syndrome. There was no correlation between FM% and biochemical variables.

### 3.2. Flow Cytometry Analysis

Statistically significant differences were observed in the percentage of CD4^+^ T lymphocytes only in the individuals diagnosed with hypertension when compared to the controls. A higher percentage of CD4^+^ memory T lymphocytes was also identified but the difference was statistically insignificant There were no statistically significant differences in CD8^+^ T lymphocytes, nor were they detected in CD4^+^ naïve, CD8^+^ naïve, and CD8^+^ memory T lymphocytes in patients diagnosed with hypertension. There were no differences in the CD4/CD8 ratio between hypertensive and control participants. The 61% elderly individuals with hypertension demonstrated the CD4/CD8 ratio within the reference range (≥1 or ≤2.5) and only 4% had a CD4/CD8 ratio of <1. However, 34% of hypertensive individuals had a CD4/CD8 ratio > 2.5 when compared to patients without hypertension, where 28% had CD4/CD8 > 2.5 (Table 3). The percentage of CD4^+^ T lymphocytes in hypertensive patients was also observed to depend on anthropometric parameters (BMI, FM and FMI) and gender (Figure 3). Interestingly, men with hypertension showed a similar number of CD4^+^ lymphocytes to the percentage recorder in women who had not been diagnosed with hypertension.

CD4^+^ and CD8^+^ T lymphocytes were analysed within naïve and memory subpopulations in youngest-old vs. middle-old and oldest-old age categories (Figure 4). The data showed statistically significant differences in the CD8^+^ T lymphocytes (Figure 4B). The percentage of CD8^+^ naïve T lymphocytes population (Figure 4E) and CD8^+^ memory T lymphocytes were significantly higher in the youngest-old when compared to oldest-old participants (Figure 4F). The CD4/CD8 ratio was found to be significantly higher in the 75–90 years age group (Figure 4G). There were no statistically significant differences in CD4^+^ (Figure 4A), CD4^+^ naïve (Figure 4C) and CD4^+^ memory (Figure 4D) T lymphocytes populations, as well as CD4/CD45RA/CD4CD45RO (Figure 4H) and CD8CD45RA/CD8CD45RO ratios (Figure 4I) between youngest-old vs. middle-old/oldest-old groups. An CD4/CD8 ratio < 1 was observed only in 13% of the participants aged 60–74 years. The ratio CD4/CD8 ≥ 1 or ≤ 2.5 was noted only in 33% of the participants aged 75–90 years and in 76% of the elderly falling within 60–74 years range. A ratio of CD4/CD8 > 2.5 was found in 67% of the individuals aged 75–90 years and in 11% of the participants aged 60–74 years. In the elderly aged 60–74 years with a normal BMI, the CD4/CD8 ratio < 1 was recorded in only 11% of the individuals. Most of our elderly participants aged 75–90 years with normal BMI demonstrated CD4/CD8 ≥ 1 or ≤ 2.5, while in the overweight and obese participants, the CD4/CD8 ratio within the reference range was detected in 19% and 37%, respectively. The elderly individuals (79%) aged 60–74 years with a normal BMI demonstrated the CD4/CD8 ratio within the reference range (≥1 or ≤2.5) and only 10% had the CD4/CD8 ratio > 2.5. Contrastingly, the overweight and obese individuals aged 60–74 years demonstrated the CD4/CD8 ratio < 1 at the level of 9% and 27%, respectively.

CD4^+^ and CD8^+^ T lymphocytes were also analysed within naïve and memory subpopulations between older men and women (Figure 5). A statistically significant difference was observed between older women and older men in the CD4^+^ naïve (Figure 5C) population of T lymphocytes, as well as in CD4CD45RA/CD4CD45RO (Figure 5H). The CD4/CD8 ratio tended to reach high levels in women but the values were not statistically significant (Figure 5G). The CD4/CD8 ratio < 1 was observed in 25% of men. The ratio CD4/CD8 ≥ 1 or ≤ 2.5 was recorded in 63% women and 40% men and CD4/CD8 > 2.5 was found in 32% women and 25% men.

### 3.3. CMV IgG Status and Immune Cells

The IgG CMV+ elderly participants diagnosed with hypertension manifested higher values of CD4^+^ when compared to the IgG CMV+ controls, which can suggest that higher values of CD4^+^ are not independent of CMV infection. (Figure 6). Ig CMV+ seniors aged 60–74 (89%) showed statistically significantly higher values of CD8^+^ when compared to IgG CMV+ seniors aged 75–90 (86%) (Figure 7A). CD8+ naïve T lymphocytes cells were significantly higher in IgG CMV+ 60–74 group when compared to 75–90 IgG CMV+ elderly (Figure 7B) Higher values of CD4/CD8 ratio in IgG CMV+ 75–90 years old seniors were also noted (Figure 7C). In elderly groups, the majority of women were IgG CMV+ (90%), while seropositivity of CMV was diagnosed in 75% of men. Interestingly, statistically significant differences were found between IgG CMV+ men and IgG CMV+ women in CD4^+^ naïve T lymphocytes cells, as well as in CD4CD45RA/CD4CD45RO, which suggests a relationship between the seropositivity of CMV and gender (Figure 7D,E). Statistically significant differences were identified between CD4^+^ naïve T lymphocytes in IgG CMV+ and IgG CMV− men (Figure 6). The value *η*^2^ indicated a large effect of age and CMV infection on the result of the CD8^+^, as well as CD4/CD8, and also a large effect of sex and CMV infection on the result of CD4^+^ naïve T lymphocytes.

### 3.4. Haematological Variables

The white blood cells count, as well as the parameters of the red blood cells, such as RBC, HB and HCT, MCV, MCH, and MCHC, were in the range of referential values in all participants (Table 4). The platelet count was higher in patients diagnosed with hypertension when compared to the control group, however the references were not statistically significant.

### 3.5. Biochemical Variables

Lipoproteins and glucose have been proven to be the strongest biomarkers of aging. High levels of TC > 200 mg/dL, LDL > 130 mg/dL and non-HDL > 130 mg/dL were found in approximately 60% of elderly diagnosed with hypertension and in 50% of the control group. However, the lipoprotein–lipid profile, including oxLDL, and glucose did not differ between hypertension and controls (Table 5). In our study, CRP concentrations were found at higher values only in *n* = 4 elderly individuals. There were no statistically significant differences in the CRP concentrations between both groups. However, *η^2^* analysis showed no immune aging nor disease effect.

## 4. Discussion

Age is one of the major factors that affects the development of many diseases, and thus, age-related diseases are becoming an increasing public health problem. The reported high morbidity and mortality of the elderly caused or triggered by influenza viruses, despite existing vaccination programs, is an ideal example to illustrate the defectiveness of the adaptive immune system. Due to the lack of previously developed immunity, the ongoing COVID-19 pandemic has emphasized even more drastically how defective primary immune responses can get with advancing age [24].

The elderly with hypertension were the most numerous groups tested in our study. According to Yu & Shin [25], T cell senescence is related to cardiovascular diseases (CVDs) such as atherosclerosis, acute myocardial infarction, and hypertension. In the study conducted to determine the T lymphocyte phenotype in newly diagnosed hypertensive patients, Youn et al. [26] showed a significantly higher number of circulating immunosenescent pro-inflammatory CD8^+^ T lymphocytes in 71 individuals with hypertension aged 51.6 ± 11.2 years when compared to 71 healthy ones aged 51.5 ± 12.2 years. Interestingly, in our study, we observed higher numbers of the CD4^+^ T lymphocyte in individuals with hypertension aged 72.3 ± 5.9 years when compared to the controls. Ni et al. [27] analysed 40 individuals with essential hypertension (EHs) aged: 56.14 ± 2.19 years and 40 normotensive healthy participants (NTs) aged: 53.60 ± 3.45 years and also observed a higher percentage of CD3^+^ CD4^+^ in EHs than in NTs patients. In the available literature, flow cytometry analyses have already revealed an increased infiltration of leukocytes (CD45^+^) and CD4^+^ lymphocytes in response to the infusion of angiotensin. We also observed an increase in CD4^+^ memory T lymphocytes in hypertensive participants. Itani et al. [28] showed higher values of circulating CD4^+^ and CD8^+^ memory T lymphocytes in 20 hypertensive patients aged 52.6 ± 11 years when compared to normotensive control aged 52.6 ± 12 years. The results may suggest an important role of T lymphocytes in the development of hypertension in different age groups, and could be used as therapeutic targets in this widespread disease in the future. According to Alonso-Fernandez & De la Fuenta [29], several age-related changes in immune functions can be linked to longevity and the predictors of mortality include increased IL-6 levels and the ratio CD4/CD8 < 1. In our study, the CD4/CD8 ratio > 2.5 was observed in approximately 34% of the participants with hypertension, and only 4% had CD4/CD8 < 1. Ni et al. [27] also noted a higher CD4/CD8 ratio, as well as serum IFN-γ and TNF-α levels, in essential hypertension when compared to normotensive individuals. Men in their 50 s are at a greater risk of developing high blood pressure than women; however, in women, due to the decline of the level of estrogens, the risk of developing the disease increases in the menopausal period [30,31]. Our analyses revealed that the percentage of CD4^+^ in men with hypertension was at a similar level as in women without hypertension. It has been proven that, due to their greater susceptibility to autoimmune diseases, women have more circulating CD4^+^ T lymphocytes than men. In addition, CD4^+^ T cells in women are more likely to expand in response to antigenic stimulation and produce higher levels of Th1 when compared to men, whose T cells are more biased towards Th17 cytokine production. The available research outcomes have suggested that the individuals with CMV infection run an increased risk of hypertension, however, the effect of CMV infection on blood pressure is still equivocal [32]. Our research showed that the IgG CMV+ elderly diagnosed with hypertension achieved higher values of CD4^+^ T cells when compared to the controls, which may indicate that the increase in CD4^+^ T cells in hypertensive patients is independent of CMV infection. Nevertheless, due to the limited number of groups, the study must be continued.

The process of ageing has an impact on both naïve CD4^+^ T and CD8^+^ T lymphocytes, yet they are affected in slightly different ways. The number of CD4^+^ naïve T lymphocytes is stable for most of the lifespan until around the age of 70 years, when a drastic decline and reduction of their repertoire is observed. In contrast to CD4^+^ naïve T lymphocytes, CD8^+^ naïve T cell counts appear to be more susceptible to death receptor-mediated apoptosis, tumour necrosis factor (TNFα), or Fas ligand, and is therefore more sensitive to age-related changes [33]. Consistently, we observed a reduced percentage of CD8CD45RA lymphocytes in seniors aged 75–90 when compared to seniors aged 60–74 years. Similar observations were reported by Gupta et al. [34], who additionally showed that the decrease in CD8^+^ naïve T lymphocytes in the elderly was a consequence of an increased activation of caspase 8 and caspase 3. According to Goronzy et al. [35], in most elderly individuals, the proportion of CD4^+^ naïve T lymphocytes remains constant until around 75 years of age and only later on does it decline more rapidly. Interestingly, our research in the Polish population did not confirm these observations. We showed that the percentage of CD4^+^ naïve T lymphocytes increased in seniors aged 75–90 when compared to seniors aged 60–74. The described changes may result from different antigenic stimulation during life and thus a lower resistance to telomerase induction. This is yet another phenomenon which requires further research [36].

CMV is a major driving antigen for the replicative senescence of T cells and, thus, it affects the CD4/CD8 ratio and it is part of the original “Immune Risk Profile” (IRP) [25,37]. Essentially, the IRP could be defined using solely an CD4/CD8 < 1 as a surrogate marker, presented in about 15% of free-living 85-years-olds. In our study 60% of the elderly demonstrated the CD4/CD8 ratio of ≥1 or ≤2.5, while the CD4/CD8 ratio > 2.5 was identified in 32% of older ones. Interestingly, we demonstrated the CD4/CD8 ratio < 1 only in the 60–74 age group, but the CD4/CD8 ratio > 2.5 was identified in 67% of 75–90-year-old individuals. In turn, in the Leiden 85+ study, only 2% of the individuals aged 89 years showed an CD4/CD8 ratio of <1, compared to 20% in those between 70 and 81 years of age. Vasson et al. [10] also analysed the biomarkers of immune status in 300 healthy volunteers aged between 20–75 years, recruited in Austria, Spain, and France, and they compared three European countries and changes in T, B, and NK cells. The immune parameters in French volunteers were recorded as constantly intermediate values in comparison with those measured in Austrian and Spanish individuals, thus reflecting the importance of a geographic variability on the immune status and possible gradual differences in the dietary intake, lifestyle habits, environmental, genetic, and socio-economic factors between Southern and Northern Europe. In addition, they also noticed changes in the CD4/CD8 ratio which was significantly different in Austria (1.8 ± 0.1), in France (1.5 ± 0.1), and in Spain (1.2 ± 0.1). In turn, in the Polish population, we showed the statistically significant differences in CD4/CD8 ratio in the IgG CMV+ elderly aged 60–74 years in comparison to the IgG CMV+ elderly aged 75–90 Interestingly, our research findings contradict the results reported by Adriaensen et al. [37], in which a decrease in the CD4/CD8 ratio was observed in patients with IgG CMV+. The differences between countries are not only related to factors such as eating or social habits, but also to daily physical activity, as shown in our previous study [16].

Another retrospective population-based study, the Kristianstad Survey (KRIS), together with the OCTO and NONA studies, showed that the proportion of individuals with an ratio CD4/CD8 < 1 was significantly higher in men when compared to women [38]. Hirokawa et al. [39] also reported that the CD4/CD8 of <1 concerned mainly men, while the number of CD8^+^ naïve T lymphocytes and the CD4/CD8 ratio were significantly higher in women. Research carried out in Poles aged 65–74 years has shown that twice as many males (over 20%) than females (10%) have a CD4/CD8 ratio equal or lower than one [40]. In our study a significantly higher percentage of CD4^+^ naïve T lymphocytes was observed in women when compared to men. The majority of the participants (83%) were females, with only 16% being males and yet, approximately 25% of the male group showed a CD4/CD8 ratio of <1. It has been proven that the disproportion in the CD4/CD8 ratio is not only a consequence of atrophy of the thymus gland, but it is also affected by hormonal changes. The effect of hormones on the CD4/CD8 ratio was confirmed by the correlation between low plasma estradiol levels, high CD8^+^ T lymphocyte values, and a low CD4/CD8 ratio [19]. The gender disproportion in the CD4^+^ naïve T cell population can also depend on CMV infection. We observed that IgG CMV+ men showed a statistically significantly lower number of CD4^+^ naïve T cells when compared to IgG CMV− men. Curiously, IgG CMV+ women showed a statistically significantly higher number of CD4^+^ naïve T cells when compared to IgG CMV+ men. It might be assumed that CMV reactivation in men and women manifests itself differently, but this has not yet been fully explained. Nonetheless, the observed gender-related trend of changes in T lymphocyte populations does not suffice to draw unequivocal conclusions, due to a small sample size. To provide more substantial evidence, the next stage of the study should include a larger male sample size.

Life expectancy has increased significantly over the past decades. The disproportions in the cells of the innate and adaptive responses, as well as chronic low-grade inflammation, contribute to the development of hypertension, neurodegenerative, metabolic, and neoplastic diseases. The identification of disease-specific signalling pathways in old age that regulate the immune response is an important milestone in defining the role of immunosenescence in chronic diseases. Understanding the exact relationship between immunosenescence and chronic inflammation can greatly improve the prevention and treatment of age-related diseases [41].

## 5. Conclusions

The present study showed increased CD4^+^ T lymphocytes in individuals with elevated blood pressure, which emphasizes the role of the immune system in the development of arterial hypertension. CMV infection reduces the percentage of CD4^+^ naïve T cells, which enhances hypertension development, especially in older men. The CD4/CD8 ratio is influenced not only by comorbidities, but also by age and gender. It has been shown that the value of CD4/CD8 < 1 was dominant in men and in the 60–74 years age group. This indicates that the CD4/CD8 ratio included in IRP can be helpful in CVD risk assessment in older people; however, it needs to be validated in large cohort studies with equal proportions of genders.

## 6. Limitations

The limitations of the study include a relatively small number of participants, an especially unequal proportion of genders, and no information on their lifestyle and environmental factors.

## Figures and Tables

**Figure 1 jcm-11-00291-f001:**
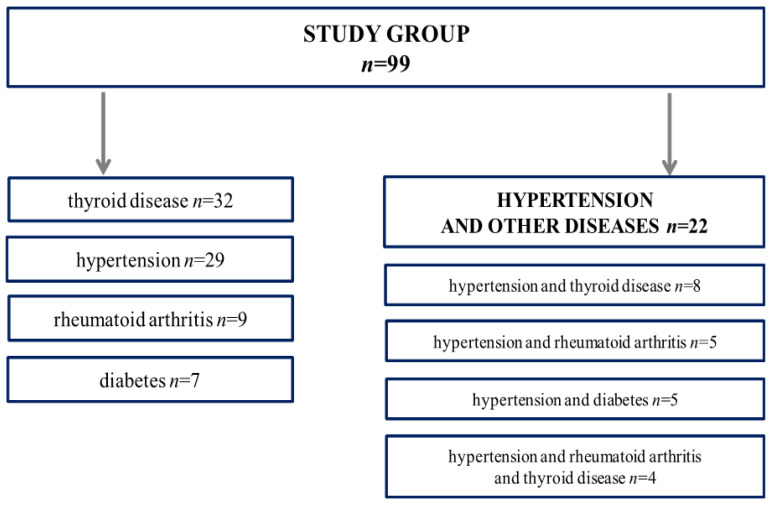
Schematic illustration of the participants’ selection according to age-related diseases.

**Figure 2 jcm-11-00291-f002:**
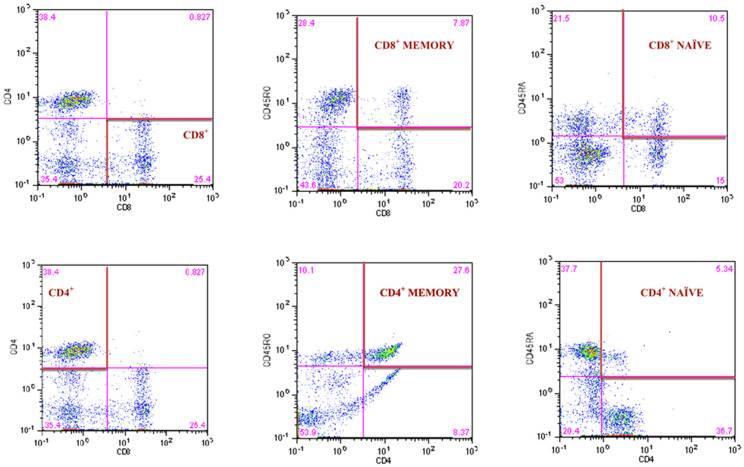
Gating strategy for identifying the CD4^+^ and CD8^+^ T lymphocyte and the frequency of CD4^+^ and CD8^+^ naïve and memory T lymphocytes.

**Figure 3 jcm-11-00291-f003:**
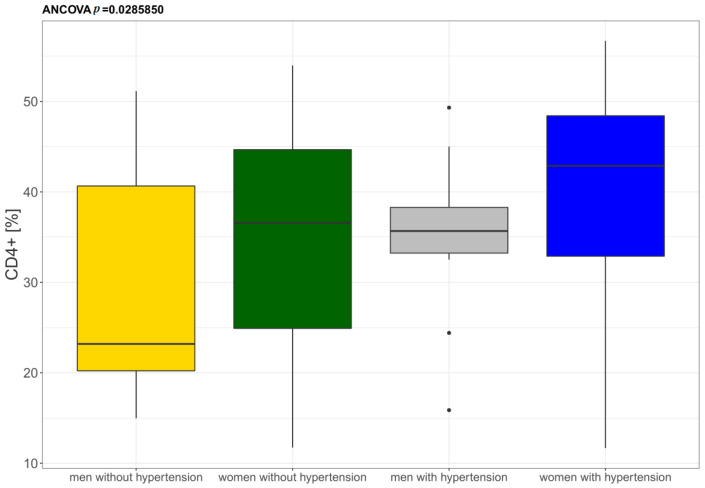
Differentiation of the percentage of total CD4^+^ in women and men diagnosed with hypertension (*n* = 51) and without hypertension (*n* = 48).

**Figure 4 jcm-11-00291-f004:**
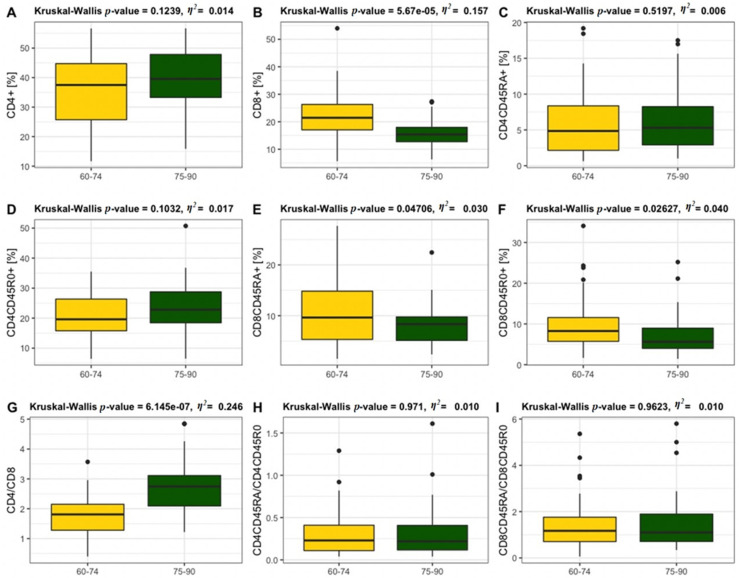
Percentages of total CD4^+^ (**A**) and CD8^+^ (**B**) T lymphocytes and the CD4^+^ naïve (**C**) and CD4^+^ memory (**D**) and CD8^+^ naïve (**E**) and CD8^+^ memory (**F**) T lymphocytes in 60–74 years (*n* = 63) of age and 75–90 (*n* = 36) years of age and the following ratios: CD4/CD8 (**G**), CD4CD45RA/CD4CD45RO (**H**) and CD8CD45RA/CD8CD45RO (**I**).

**Figure 5 jcm-11-00291-f005:**
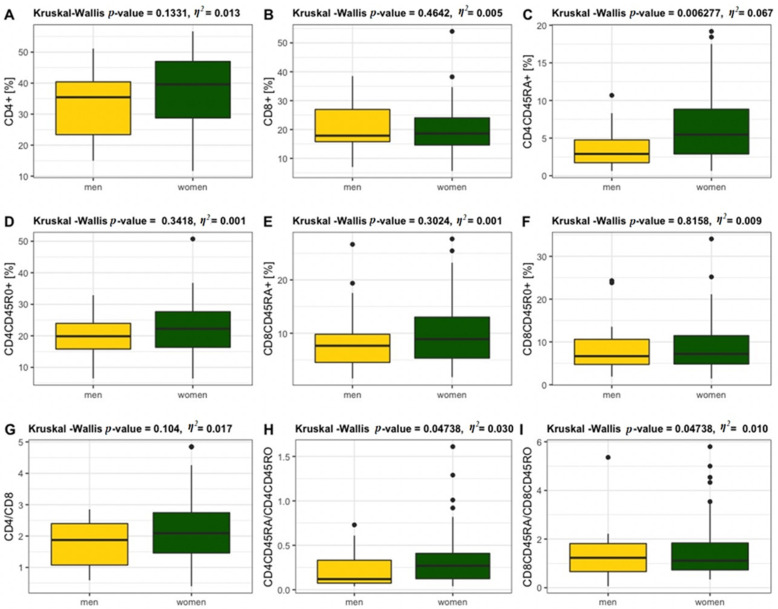
Percentages of total CD4 ^+^ (**A**) and CD8^+^ (**B**) T lymphocytes and the CD4^+^ naïve (**C**) and CD4^+^ memory (**D**) and CD8^+^ naïve (**E**) and CD8^+^ memory (**F**) T lymphocyte in older women (*n* = 83) and men (*n* = 16) and the following ratios: CD4/CD8 (**G**), CD4CD45RA/CD4CD45RO (**H**) and CD8CD45RA/CD8CD45RO (**I**).

**Figure 6 jcm-11-00291-f006:**
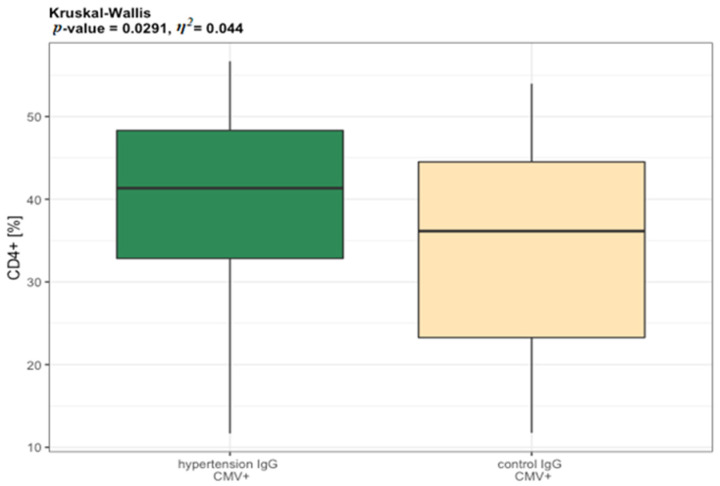
Distribution of T lymphocytes phenotypes in relation to CMV serostatus. Differences between hypertension (*n* = 44) vs. controls (*n* = 43). Abbreviations: CMV+ cytomegalovirus positive. The measurements in groups were compared by the one-way ANOVA or the Kruskal-Wallis non-parametric test (if the normality assumption is violated). *η^2^* is a measure of effect size.

**Figure 7 jcm-11-00291-f007:**
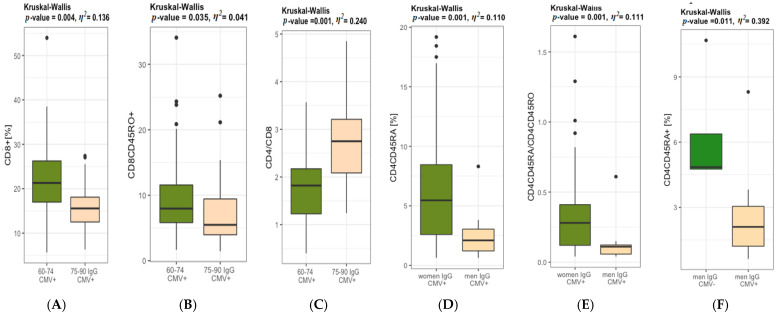
Distribution of T lymphocytes phenotypes in relation to CMV serostatus. Differences between 60–74 years of age (*n* = 56) vs. 75–90 years of age (*n* = 31) (**A**–**C**), older women (*n* = 75) vs. older men (*n* = 12) (**D**,**E**), older men IgG CMV− (*n* = 4) vs. older men IgG CMV+ (*n* = 12) (**F**). Abbreviations: CMV+ cytomegalovirus positive, CMV− cytomegalovirus negative. The measurements in groups were compared by the one-way ANOVA or the Kruskal-Wallis non-parametric test (if the normality assumption is violated), *η*^2^ is a measure of effect size, SD standard deviation, Me median.

**Table 1 jcm-11-00291-t001:** Anthropometrics and body composition.

	Hypertension *n* = 51	Control *n* = 48	Hypertension vs. Control *p*-Value	*η* ^2^
	Mean ± SD (Me)	Mean ± SD (Me)		
Age (years)	72.3 ± 5.9 (72.0)	70.2 ±5.5 (70.0)	0.152	0.010
Weight (kg)	72.3 ± 10.6 (70.1)	67.1± 9.7 (68.0)	0.014	0.061
Height (cm)	162.1 ± 6.9 (162.1)	159.4 ± 4.9 (159.0)	0.038	0.034
BMI (kg/m^2^)	27.6 ±3.6 (27.1)	26.4 ± 3.4 (25.8)	0.090	0.019
BMI 18.5–24.9 kg/m^2^ (%)	24.0	38.8		
BMI 25–29.9 kg/m^2^ (%)	56.0	42.8		
BMI ≥ 30 kg/m^2^ (%)	20.0	18.4		
MM (kg)	44.7 ± 7.8 (4264)	42.2 ± 6.0 (40.9)	0.083	0.034
FFM (kg)	47.5 ± 8.1 (45.4)	44.2 ± 6.2 (43.5)	0.035	0.035
FFMI (kg/m^2^)	18.0 ± 2.3 (17.6)	17.4 ± 1.9 (16.9)	0.118	0.015
FM (kg)	24.8 ± 5.6 (24.7)	22.9 ± 5.8 (22.8)	0.089	0.029
FM%	34.3 ± 5.6 (35.3)	33.8 ± 5.2 (34.4)	0.099	0.028
FMI (kg/m^2^)	9.5 ± 2.4 (9.5)	9.0 ± 2.3 (8.8)	0.263	0.007
SBP (mmHg)	150.9 ± 18.5 (149.5)	142.1 ± 18.6 (144.0)	0.022	0.054
DBP (mmHg)	81.2 ± 11.9 (82.0)	82.6 ± 15.8 (78.0)	0.945	0.010

Abbreviations: BMI body mass index, MM muscle mass, FFM fat-free mass, FFMI fat-free mass index, FM fat mass, FMI fat mass index. SBP systolic blood pressure DBP diastolic blood pressure. The measurements in groups were compared by the one-way ANOVA or the Kruskal-Wallis non-parametric test (if the normality assumption is violated), *η*^2^ is a measure of effect size, SD standard deviation, Me median.

**Table 2 jcm-11-00291-t002:** Anthropometrics and body composition differences between 60–74 years and 75–90 years as well as between women and men.

	60–74 Years *n* = 63	75–90 Years *n* = 36	60–74 Years vs. 75–90 Years *p*-Value	Females *n* = 83	Males *n* = 16	Females vs. Males *p*-Value
	Mean ± SD (Me)	Mean ± SD (Me)		Mean ± SD (Me)	Mean ± SD (Me)	
Age (years)	67.9 ± 3.5 (68.0)	77.7 ±3.5 (76.5)	0.001	70.8 ± 5.8 (70.0)	73.6 ± 5.5 (72.0)	0.087
Weight (kg)	70.1 ± 9.7 (68.7)	69.1± 12.1 (68.5)	0.100	68.6 ± 9.5 (68.1)	75.9 ± 13.4 (74.5)	0.009
Height (cm)	161.5 ± 5.9 (161.0)	159.4 ± 6.4 (159.0)	0.642	159.5 ± 4.9 (159.0)	167.3 ± 7.7 (169.5)	0.001
BMI (kg/m^2^)	26.9 ± 3.1 (26.7)	27.3 ± 4.3 (26.7)	0.754	27.0 ± 3.6 (26.7)	27.2 ± 3.6 (26.6)	0.665
MM (kg)	43.8 ± 6.9 (42.2)	33.9 ± 5.5 (34.7)	0.319	41.0 ± 3.7 (41.0)	57.6 ± 4.5 (56.0)	0.001
FFM (kg)	46.3 ± 7.5 (44.5)	45.1 ± 7.4 (44.1)	0.546	44.2 ± 5.1 (44.3)	54.6 ± 11.2 (57.9)	0.001
FFMI (kg/m^2^)	17.2 ± 2.0 (17.2)	17.7 ± 2.4 (17.4)	0.785	17.4 ± 1.8 (17.1)	19.4 ± 3.0 (20.2)	0.004
FM (kg)	23.8 ± 5.4 (23.3)	24.0 ± 6.7 (22.6)	0.870	24.4 ± 5.6 (23.4)	21.3 ± 6.3 (20.0)	0.055
FM%	33.9 ± 5.5 (34.7)	34.3 ± 5.3 (35.0)	0.683	35.2 ± 4.2 (35.1)	28.2 ± 7.2 (27.7)	0.001
FMI (kg/m^2^)	9.2 ± 2.2 (9.2)	9.5 ± 2.7 (9.4)	0.525	9.6 ± 2.2 (9.4)	7.7 ± 2.5 (7.0)	0.003
SBP (mmHg)	143.2 ± 18.7 (143.5)	153.2 ± 17.9 (151.0)	0.013	146.3 ± 18.7 (147.0)	148.3 ± 20.8 (147.0)	0.713
DBP (mmHg)	82.6 ± 14.8 (78.5)	80.8 ± 10.7 (82.0)	0.816	81.7 ± 13.7 (79.0)	83.7 ± 12.5 (83.0)	0.500

Abbreviations: BMI body mass index, MM muscle mass, FFM fat-free mass, FFMI fat-free mass index, FM fat mass, FMI fat mass index. SBP systolic blood pressure DBP diastolic blood pressure. The measurements in groups were compared by the one-way ANOVA or the Kruskal-Wallis non-parametric test (if the normality assumption is violated), SD standard deviation, Me median.

**Table 3 jcm-11-00291-t003:** Distribution of T lymphocytes in patients diagnosed with hypertension when compared to controls.

T Lymphocytes	Hypertension *n* = 51	Control *n* = 48	Hypertension vs. Control *p*-Value	*η* ^2^
(%)	Mean ± SD (Me)	Mean ± SD (Me)		
CD4^+^	39.4 ± 10.7 (40.2)	34.3 ± 11.7 (36.4)	0.041	0.033
CD8^+^	19.9 ± 6.5 (18.4)	19.2 ± 9.3 (18.9)	0.446	0.004
CD4CD45RA^+^	6.3 ± 4.1 (5.3)	5.7 ± 4.5 (4.1)	0.245	0.003
CD4CD45RO^+^	22.7 ± 7.3 (22.7)	20.4 ± 8.2 (19.3)	0.086	0.020
CD8CD45RA^+^	9.4 ± 5.0 (8.6)	10.3 ± 6.7 (8.7)	0.897	0.010
CD8CD45RO^+^	8.9 ± 5.1 (7.5)	8.8 ± 6.4 (6.5)	0.456	0.005
CD4/CD8	2.2 ± 0.9 (2.0)	2.1 ± 0.9 (2.1)	0.766	0.009
<1	4.0	12.2		
≥1 or ≤2.5	62.0	59.2		
>2.5	34.0	28.6		
CD4CD45RA/CD4CD45RO	0.3 ± 0.3 (0.2)	0.3 ± 0.2 (0.2)	0.698	0.009
CD8CD45RA/CD8CD45RO	1.4 ± 1.1 (1.1)	1.5 ± 1.1 (1.2)	0.437	0.004

The measurements in groups were compared by the one-way ANOVA or the Kruskal-Wallis non-parametric test (if the normality assumption is violated). *η*^2^ is a measure of effect size, SD standard deviation, Me median.

**Table 4 jcm-11-00291-t004:** Haematological variables (mean ± SD).

	Reference Values	Hypertension *n* = 51	Control *n* = 48	Hypertension vs. Control *p*-Value	*η* ^2^
		Mean ± SD (Me)	Mean ± SD (Me)		
Leukocytes (10^3^/µL)	5.0–11.6	6.9 ± 2.0 (6.6)	6.2 ± 1.5 (6.1)	0.081	0.021
Lymphocytes (10^3^/µL)	1.3–4.0	2.2 ± 0.7 (2.2)	2.2 ± 0.7 (2.1)	0.351	0.001
Granulocytes (10^3^/µL)	2.4–7.6	4.2 ± 1.6 (3.9)	3.6 ± 1.2 (3.5)	0.124	0.014
LYM%	19.1–48.5	33.3 ± 9.0 (33.2)	35.2 ± 9.0 (35.0)	0.303	0.002
GRA%	43.6–73.4	59.3 ± 9.5 (58.7)	56.4 ± 10.1 (56.6)	0.153	0.014
RBC (10^3^/µL)	F 4.0–5.5 M 4.5–6.6	4.8 ± 0.3 (4.8)	4.8 ± 0.3 (4.8)	0.916	0.000
HB (g/dL)	F 12.5–16.0 M 13.5–18.0	13.8 ± 0.7 (13.7)	13.9 ± 0.8 (13.9)	0.439	0.006
HCT (%)	F 37–47 M 40.0–51.0	39.4 ± 2.3 (39.1)	39.8 ± 2.4 (39.5)	0.470	0.005
MCV (fL)	F 80–95 M 80–97	81.6 ± 2.5 (82.0)	82.3 ± 3.5 (82.0)	0.346	0.001
MCH (pg)	F 27.0–32.0 M 26.0–32.0	28.6 ± 1.0 (28.5)	28.8 ± 1.4 28.7)	0.412	0.007
MCHC (g/dL)	F 32.0–36.0 M 31.0–36.0	35.0 ± 0.8 (35.2)	35.0 ± 0.7 (35.0)	0.604	0.007
PLT (10^3^/µL)	150–400	265.9 ± 57.4 (257.5)	236.8 ± 65.1(247.0)	0.293	0.001

Abbreviations: LYM lymphocytes, GRA granulocytes, RBC red blood cells, HB haemoglobin, HCT haematocrit, MCV mean corpuscular volume, MCH mean cells haemoglobin, MCHC mean corpuscular/haemoglobin concentration, PLT platelets, F female, M male. The measurements in groups were compared by the one-way ANOVA or the Kruskal-Wallis non-parametric test (if the normality assumption is violated), *η*^2^ is a measure of effect size, SD standard deviation, Me median.

**Table 5 jcm-11-00291-t005:** Biochemical variables (mean ± SD).

	Reference Values	Hypertension *n* = 51	Control *n* = 48	Hypertension vs. Control *p*-Value	*η^2^*
		Mean ± SD (Me)	Mean ± SD (Me)		
Glucose (mg/dL)	60–115	96.6 ± 13.7 (93.0)	95.4 ± 13.5 (92.8)	0.592	0.007
TC (mg/dL)	<200	237.6 ± 56.3 (234.0)	250.4 ± 50.1 (245.0)	0.240	0.014
TG (mg/dL)	<150	120.3 ± 51.5 (115.8)	124.5 ± 65.1 (119.5)	0.933	0.010
HDL (mg/dL)	desirable >60	80.1 ± 15.8 (80.1)	79.3 ± 12.0 (81.0)	0.980	0.010
LDL (mg/dL)	<130	129.7 ± 49.8 (125.3)	140.1 ± 41.7 (136.8)	0.267	0.013
non-HDL (mg/dL)	<130	157.5 ± 60.7 (60.7)	171.1 ± 48.1 (165.9)	0.226	0.006
oxLDL	-	413.5 ± 424.2 (127.9)	531.7 ± 455.8 (381.8)	0.567	0.007
CRP (mg/L)	0.068–8.2	2.6 ± 2.4 (1.9)	2.6 ± 2.4 (1.9)	0.972	0.010

Abbreviations: TG triglycerides, TC total cholesterol, LDL low density lipoprotein, HDL high density lipoprotein, oxLDL oxidized low-density lipoprotein, CRP C-reactive protein. The measurements in groups were compared by the one-way ANOVA or the Kruskal-Wallis non-parametric test (if the normality assumption is violated), *η^2^* is a measure of effect size, SD standard deviation, Me median.

## Data Availability

The data used to support the findings of this study are available from the corresponding author upon request.

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
