# Peer review of "Pre-Existing Hypertension Is Related with Disproportions in T-Lymphocytes in Older Age"

_jcm, 2022, doi:10.3390/jcm11020291_

Round 1
Reviewer 1 Report
Summary
In this manuscript, Tylutka et al, have evaluated the immunosenescence pattern of naïve and memory T cells subpopulations and the immune risk profile (IRP) expressed as CD4/CD8 ratio and IgG CMV related to comorbidities by using flow cytometry. They showed an increased CD4+ T lymphocytes in patients diagnosed with hypertension compared to control group, which emphasizes the role of the immune system in the development of arterial hypertension. They also reported that the analyzed CD4/CD8 ratio are also dependent on age and gender.
Importantly, they identified the changes in naïve and memory T lymphocyte population, CD4/CD8 and CMV seropositivity included in IRP as important markers of health status in the elderly dependent on hypertension.
The study is well defined, and the experiments are performed accurately. The results of the study well support the conclusions and hence the study is significant. However, there are certain issues in this current manuscript which requires to be addressed for the editor’s consideration.
Major comments
- CMV is a major driving antigen for the replicative senescence of T cells hence, the authors should consider evaluating CMV-specific IFN-γ or TNF-α secretion or the cytotoxic function of CD8+ T cells as they are positively correlated with PWV in multivariate analysis as shown in Yu HT, Shin EC. T cell immunosenescence, hypertension, and arterial stiffness. Epidemiol Health. 2014;36:e2014005. Published 2014 May 23. doi:10.4178/epih/e2014005
- CMV is a major factor accelerating T cell senescence. T cell senescence is related to cardiovascular diseases. CMV seropositivity included in IRP can be a helpful marker of health status in older adults, the authors understand that the analysis needs to be expanded to include information on the lifestyle and environmental factors. However, they should also include the cardiovascular parameters as the presence of these senescent T cells also correlates with the occurrence of a first cardiovascular event and with worse disease outcomes. The roles of T cell senescence and CMV infection in cardiovascular disease need to be validated in cohort studies, and the mechanism by which senescent T cells contribute to the pathogenesis of cardiovascular disease requires further investigation
Minor comments
- The authors observed higher values of both circulating CD4+ and CD8+ memory T lymphocytes in individuals with elevated blood pressure. Thus, these findings suggest that CD4+ T cells can serve as markers and biomarkers. However, the discussion should be described as “Both and CD8+ memory T lymphocytes” can serve as markers in the development of hypertension and might be therapeutic targets for this widespread diseases.
- The study faces several limitations as mentioned in the limitation section, “There are relatively small number of participants, There is unequal ration of gender. Also further information on their lifestyle and environmental factors will be determining factors in deriving further conclusions.
Author Response
Response to Review 1
We greatly appreciate your time and effort dedicated to providing feedback on our manuscript and we are grateful for the insightful comments on and valuable improvements to our paper. All the suggestions helped us to evaluate our outcomes even more precisely in order to deliver an improved, high quality scientific manuscript which we hope will now meet the high standards of Journal of Clinical Medicine.
Major comments
- CMV is a major driving antigen for the replicative senescence of T cells hence, the authors should consider evaluating CMV-specific IFN-γ or TNF-α secretion or the cytotoxic function of CD8+ T cells as they are positively correlated with PWV in multivariate analysis as shown in Yu HT, Shin EC. T cell immunosenescence, hypertension, and arterial stiffness. Epidemiol Health. 2014;36:e2014005. Published 2014 May 23. doi:10.4178/epih/e2014005
We agree with the Reviewer’s suggestion. The study is still being continued involving the assessment of IFNγ, TNFα and other inflammatory molecules related to the function of lymphocytes T. The paper by Yu & Shin Epidemiol Health 2014 has been referred to in the manuscript (1st and 4th paragraph in Discussion).
- CMV is a major factor accelerating T cell senescence. T cell senescence is related to cardiovascular diseases. CMV seropositivity included in IRP can be a helpful marker of health status in older adults, the authors understand that the analysis needs to be expanded to include information on the lifestyle and environmental factors. However, they should also include the cardiovascular parameters as the presence of these senescent T cells also correlates with the occurrence of a first cardiovascular event and with worse disease outcomes. The roles of T cell senescence and CMV infection in cardiovascular disease need to be validated in cohort studies, and the mechanism by which senescent T cells contribute to the pathogenesis of cardiovascular disease requires further investigation.
Thank you for a valuable remark. This comment has been included in Conclusion.
Minor comments
- The authors observed higher values of both circulating CD4+ and CD8+ memory T lymphocytes in individuals with elevated blood pressure. Thus, these findings suggest that CD4+ T cells can serve as markers and biomarkers. However, the discussion should be described as “Both and CD8+ memory T lymphocytes” can serve as markers in the development of hypertension and might be therapeutic targets for these widespread diseases.
Discussion and Conclusion have been significantly revised according to the observed results.
- The study faces several limitations as mentioned in the limitation section, “There are relatively small number of participants, there is unequal ration of gender. Also, further information on their lifestyle and environmental factors will be determining factors in deriving further conclusions.
We appreciate the Reviewer’s comment which has been included in Limitations.

Reviewer 2 Report
Major comments
The author showed differences between immune cells in the young and identified the elderly age-related changes in the immune system. This study leads us to understand the exact relationship between immunosenescence and chronic inflammation and be able to greatly improve the prevention and treatment of age-related diseases. However, there are several critical concerns in this manuscript.
- Flow cytometry analysis
- P4, L126, 127. I feel the inverted CD4/CD8 ratio is ambiguous. I’m confused about whether the inverted CD4/CD8 means CD8/CD4 ratio or not. Or does it mean out of range in the reference range (CD4/CD8 ratio < 1 and CD4/CD8 ratio> 2.5).
- Statistical analysis
- P5, L161. Why did you use one-way ANOVA or Kruskal-Wallis which usually was used for the comparison of over 3 groups but not Student's T-test or Mann Whitney U-test which was commonly used for the that of 2 groups? Is the result such as statistical significance in your manuscript the same, even though Student's T-test or Mann Whitney U-test is used?
- P5, L162. Was eta-squared calculated for only the Kruskal-Wallis test? When one-way ANOVA is used, eta-squared wasn’t calculated?
- P5, L162. Could you describe the formula of eta-squared which was used in your manuscript? I’m wondering whether eta-squared can appropriately evaluate a measure of effect size for non-parametric or non-heteroscedasticity data.
- Did one-way ANOVA or the Kruskal-Walls non-parametric test was performed for each measurement according to the result of the parametric test? If the comparison test differs among measurements, all significances of measurement comparisons can’t be equally evaluated due to the difference in statistical power.
- Body composition
- P5, L178, Table 1. Is there non-hypertension who has high BMI and FMI like hypertension? Even though BMI and FWI were matched between populations, the result of other analyses can be expected like the result in your manuscript. Moreover, the height of the control population is smaller than that of hypertension. Furthermore, although FFMI is significantly different between hypertension and control population, what component is there in FFMI. For example, is the muscle mass difference between populations? I concern the above inhomogeneities might affect other results such as flow cytometry, CMV IgG status, and immune cells, hematological variables, biochemical variables.
- Flow cytometry analysis
- P7, L226, Figure 3. Could you show the demographic data of younger (60-74 years old) and elderly (75-90 years old) age groups like Table 1. I’m concerned about the non-uniform proportion of hypertension, high BMI, FMI, and sex between younger (60-74 years old) and elderly (75-90 years old) age groups might give the result the bias. Is there any problem with it?
- P8, L240. Could you show the demographic data of men and women groups like Table 1. I’m concerned about the non-uniform proportion of hypertension, high BMI, FMI, and age between sex groups, which might give the result a bias. Is there any problem with it?
- P7, L221. “The elderly individuals (79%) aged 60-74 years with normal BMI demonstrated the CD4/CD8 ratio within the reference range (≥1 or ≤2.5) and only 10% had the CD4/CD8 ratio >2.5.” Why part of elderly individuals has an out-of-reference range (CD4/CD8 ratio <1 or >2.5), even though their BMI is normal? Is there any characteristic difference between the elderly population within and out of the reference range?
- Discussion
- P12, L337, “Our research showed that the IgG CMV+ elderly 337 diagnosed with hypertension achieved higher values of CD4+ T cells compared to the con-338 trols, which may indicate that the increase in CD4+ T cells in hypertensive patients is in-339 dependent of CMV infection.”. Although it seems that the result includes confounding factors such as BMI, FMI, and sex, is there no bias or problem in your result? Will a statistical test like ANCOVA which can remove confounding factors show the same result as your one in this manuscript?
- P12, L374, “The recorded differences between countries/regions can possibly be related to lifestyle factors such as habits, nutrition as well as physical and mental health status.”. Is there any evidence to support this hypothesis?
Minor comments
2. Materials and methods
2.1. Participants
- P2, L78. Could you describe the criteria for hypertension?
- P2, L94. How did you define and calculate “the repeatability”? Moreover, how many people did participate the measurement?
2.2. Body composition
- P3, L93. “7:00 and 9:00” is in the morning or afternoon? Why did you determine to measure at the time? Additionally, was the measurement performed before eating something?
- P3, L95. Which part in your manuscript did follow the methods of Tylutka et al.
- P3, L98. Is this criteria appropriate to the participants in this manuscript? For example, Asian people are relatively smaller than other people, so it seems to tend that the BMI of Asian people is underestimated. Does this effect include in your study?
2.4. Flow cytometry analysis
- P4, Figure2. What does the color mean? Additionally, could you describe the meaning of purple values in a figure.
2.5. Cytomegalovirus (CMV) IgG
- P4, L136. How population was used and CV calculated?
2.7. Biochemical variables
- P4, L136. How population was used and CV calculated?
2.8. Statistical analysis
- P5, L164, 165. “no effect if 0 ≤ η2< 0.01, a small effect if 0.01≤ η2<0.06, a moderate effect if 0.06 ≤ η2< 0.14, and a large effect if η2 ≥ 0.14.” Could you show the reference?
- P165, 167. Please italicize the symbols of spearman rank correlation coefficient rs and statistical significance p.
3. Results
3.1. Body composition
- P5, L178, Table 1. Could you describe the number of males and females, BMI average, the number of BMI normal, overweight, obese, and the number of thyroid diseases, rheumatoid arthritis, diabetes for each group in the table.
- P5, L178, Please italicize the symbols of statistical significance p.
3.2. Flow cytometry analysis
- P6, L196, Table 2. Could you show the CD4/CD8 ratio within the reference range (≥1 or ≤2.5) and out of reference range (<1 or >2.5) for hypertension and control population in the table.
- P7, L226, Figure 3. Could you explain the boxplot in detail (e.x., the meaning of box, black line in the box, error bar).
3.3. CMV IgG status and immune cells
- P9, L262, Figure 5. Please italicize the symbols of statistical significance p.
- P10, L268, Figure 6. Please italicize the symbols of statistical significance p.
3.4. Haematological variables
- P10, L281, Table 3. Please italicize the symbols of statistical significance p.
3.5. Biochemical variables
- P11, L299, Table 4. Could you show the reference source for “Reference values”? Are the reference values not dependent on age and sex and suitable for your data?
3.5. Biochemical variables
- P11, L299, Table 4. Could you show the reference source for “Reference values”? Are the reference values not dependent on age and sex and suitable for your data?
4. Discussion
- P11, L315, The reference [22-25] targeted participants around 50-60 years old. Therefore, although the participants in this manuscript are much older than those in previous studies, does the reference result apply to your result?
- P12, L327, “Thus, these findings suggest that CD4+ T cells and can serve as markers and biomarkers in the development of hypertension and might be therapeutic targets for this widespread disease.” Does this sentence apply to anyone of any age?
- P12, L330-332, “several age-related changes in immune functions can be linked to longevity and the predictors of mortality include increased IL6 levels and the ratio CD4/CD8 <1.”. As Figure 3 shows, Although CD4/CD8 ratio decreases with aging, did your result match the above previous result?
- P12, L333-334, “the same observations in patients with essential hypertension compared to normotensive individuals were reported by Ni et al. [23].” How proportion of CD4/CD8 ratio >2.5 previous study observed. As I see the reference paper written by Ni et al. [23], although it looks maximum CD4/CD8 ratio is around 2.5, did your result in patients with hypertension, “CD4/CD8 ratio >2.5 was observed in approx. 34% of the participants with hypertension”, show the same observations with the previous result?
- P13, L398, “small sample size”. Please describe the sample size for each gender to analyze the gender-related trends of changes in T lymphocytes population.
Author Response
Response to Review 2
We greatly appreciate your time and effort dedicated to providing feedback on our manuscript and we are grateful for the insightful comments on and valuable improvements to our paper. All the suggestions helped us to evaluate our outcomes even more precisely in order to deliver an improved, high quality scientific manuscript which we hope will now meet the high standards of Journal of Clinical Medicine.
Major comments
The author showed differences between immune cells in the young and identified the elderly age-related changes in the immune system. This study leads us to understand the exact relationship between immunosenescence and chronic inflammation and be able to greatly improve the prevention and treatment of age-related diseases. However, there are several critical concerns in this manuscript.
Flow cytometry analysis
P4, L126, 127. I feel the inverted CD4/CD8 ratio is ambiguous. I’m confused about whether the inverted CD4/CD8 means CD8/CD4 ratio or not. Or does it mean out of range in the reference range (CD4/CD8 ratio < 1 and CD4/CD8 ratio> 2.5).
Thank you very much for this suggestion. The reference values for the CD4/CD8 ratio between ≥1or ≤ 2.5 were adopted from McBride and Striker [2017] and Strindhall et al. [2013]. Both a reduced/low (CD4/CD8 <1) and an increased ratio (CD4/CD8> 2.5) are considered to be an immune risk phenotype. Following the Reviewer’s suggestion, the word "inverted" has been removed in the descriptions and replaced with "CD4/CD8 <1 or CD4 / CD8> 2.5” consistently throughout the manuscript.
Statistical analysis
P5, L161. Why did you use one-way ANOVA or Kruskal-Wallis which usually was used for the comparison of over 3 groups but not Student's T-test or Mann Whitney U-test which was commonly used for the that of 2 groups? Is the result such as statistical significance in your manuscript the same, even though Student's T-test or Mann Whitney U-test is used?
Thank you for pointing this out. In our statistical analysis, two tests have been performed: the parametric ANOVA test or the non-parametric Kruskal-Wallis test - depending on the results of the Shapiro-Wilk test. The results of Kruskal-Wallis are comparable to those of the non-parametric test of Mann-Whitney. Nevertheless, in the case of the Mann- Whitney test, it is not possible to perform eta-Squared in statistical analysis, which is why these tests have been selected in our analyses.
P5, L162. Was eta-squared calculated for only the Kruskal-Wallis test? When one-way ANOVA is used, eta-squared wasn’t calculated?
In our statistical analysis, two tests have been performed: the parametric ANOVA test or the non-parametric Kruskal-Wallis test - depending on the results of the Shapiro-Wilk. EtaSquared has been performed in both cases. In the case of ANOVA, the formula was: a1<- aov(RBC ~ GRUPA, data = CD34A)summary(a1)etaSquared(a1)and in the case of Kruskal-Wallis the formula was:
kruskal_effsize
(CD34A,
RBC ~ GRUPA,
ci = FALSE,
conf.level = 0.95
,ci.type = "perc",
n= 1000).
Explanation of abbreviations:
CD34A- CD34A abbreviation is the name of the entire table loaded into the R system with the group data and values of the analyzed parameters in the columns
RBC - the RBC abbreviation in the presented example stands for the value of the tested factor, here ultimately the RBC value
GRUPA - the abbreviation GRUPA is the name of the column with the analyzed groups, that is: patients with hypertension in comparison and patients without hypertension (control group)
This description has been in corporate into the section describing statistical analysis which now reads as follows: „Additionally, eta-Squared for ANOVA (η2) was used as a measure of the effect size which is indicated as having no effect if 0 ≤ η2< 0.01, a minimum effect if 0.01 ≤ η2< 0.06, a moderate effect if 0.06 ≤ η2< 0.14, and a strong effect if η2 ≥ 0.14 [Miles et al. 2001, Cohen et al.1988]. The effect size for Kruskal-Wallis test as the eta-Squared (η2) was calculated based on the function in Rstudio: kruskal_effsize and = 0.01-0.06 was a small effect, 0.061-0.14 was a moderate effect and > 0.14 was a large effect [Radovanović et al. 2020].”
P5, L162. Could you describe the formula of eta-squared which was used in your manuscript? I’m wondering whether eta-squared can appropriately evaluate a measure of effect size for non-parametric or non-heteroscedasticity data.
Analysis of eta-Squared has been used to determine the strength magnitude of the influence of a given factor or process on the change of the analyzed indicator/parameter. The use of the eta-Squared analysis has been based on the previously published manuscript byAhlberg et al. 2016: https://www.ncbi.nlm.nih.gov/pmc/articles/PMC4959048/; Hagstrom et al. 2016 https://pubmed.ncbi.nlm.nih.gov/26593858/Teixeria et al. 2019 https://www.ncbi.nlm.nih.gov/pmc/articles/PMC6561224/]. Eta-Squared analysis has been performed because it is also an important test as, unlike a significance test, the effect size is independent of the sample size. Using the R studio analysis, it is possible to evaluate eta-Squared for both ANOVA and for the Krusall-Wallis test.
FORMULA OF ETA-SQUARED FOR KRUSKAL:
kruskal_effsize(
CD34A,
RBC ~ GRUPA,
ci = FALSE,
conf.level = 0.95,
ci.type = "perc",
nboot = 1000)
and for ANOVA: a1<- aov(RBC ~ GRUPA, data = CD34A)summary(a1)etaSquared(a1)
Explanation of abbreviations
CD34A- CD34A abbreviation is the name of the entire table loaded into the R system with the group data and values of the analyzed parameters in the columns
RBC - the RBC abbreviation in the presented example stands for the value of the tested factor, here ultimately the RBC value
GRUPA - the abbreviation GRUPA is the name of the column with the analyzed groups, that is: patients with hypertension in comparison and patients without hypertension
Did one-way ANOVA or the Kruskal-Walls non-parametric test was performed for each measurement according to the result of the parametric test? If the comparison test differs among measurements, all significances of measurement comparisons can’t be equally evaluated due to the difference in statistical power.
In our statistical analysis, two tests have been performed: the parametric ANOVA test or the non-parametric Kruskal-Wallis test - depending on the results of the Shapiro-Wilk test. We totally agree with the Reviewer that both of these tests are of different power. Nevertheless, due to the analyzed patients, i.e. elderly individuals with various diseases, most of the distributions were not normal especially in the case of immune cell analyzes, hence the dominant test in our analyzes was the non-parametric Kruskal-Wallis test, which hasbeen showed in all the figures (at the figure descriptions include the top p-value and the name of the test performed).
Body composition
P5, L178, Table 1. Is there non-hypertension who has high BMI and FMI like hypertension? Even though BMI and FWI were matched between populations, the result of other analyses can be expected like the result in your manuscript. Moreover, the height of the control population is smaller than that of hypertension. Furthermore, although FFMI is significantly different between hypertension and control population, what component is there in FFMI. For example, is the muscle mass difference between populations? I concern the above in homogeneities might affect other results such as flow cytometry, CMV IgG status, and immune cells, hematological variables, biochemical variables.
According to the suggestion, we have added to Table 1 the exact division of BMI into: normal body weight, overweight and obesity in both the hypertensive and the control groups. We have prepared a new table, i.e.Table 2, with a detailed division of anthropometric parameters into the 60-74 years vs. 75-90 years groups and the genders (men vs. women). We also added muscle mass values to the Table 1 and Table 2.
Flow cytometry analysis
P7, L226, Figure 3. Could you show the demographic data of younger (60-74 years old) and elderly (75-90 years old) age groups like Table 1. I’m concerned about the non-uniform proportion of hypertension, high BMI, FMI, and sex between younger (60-74 years old) and elderly (75-90 years old) age groups might give the result the bias. Is there any problem with it?
Following the Reviewer’s suggestion, we have added a new table to the manuscript (i.e.Table 2) in which we have showed the differences in anthropometric parameters between two age groups:60-74 years vs. 75-90 years as well as between genders: men vs. women.
P8, L240. Could you show the demographic data of men and women groups like Table 1. I’m concerned about the non-uniform proportion of hypertension, high BMI, FMI, and age between sex groups, which might give the result a bias. Is there any problem with it?
According to the Reviewer’s suggestion, a new table (i.e.Table 2) has been designed and included in the manuscript to show the differences in anthropometric parameters between the age groups: 60-74 years vs. 75-90 as well as between genders: men vs. women.
P7, L221. “The elderly individuals (79%) aged 60-74 years with normal BMI demonstrated the CD4/CD8 ratio within the reference range (≥1 or ≤2.5) and only 10% had the CD4/CD8 ratio >2.5.” Why part of elderly individuals has an out-of-reference range (CD4/CD8 ratio <1 or >2.5), even though their BMI is normal? Is there any characteristic difference between the elderly population within and out of the reference range?
Obesity is a major cause of preventable deaths in the Western world, and its prevalence is rapidly increasing. So far, however, only a limited number of studies have investigated the composition of the peripheral blood immune system in obesity. Positive correlations have been reported between BMI and the total white blood cell count and T-cell numbers in peripheral blood, but conflicting data have been published as well. In the peripheral blood T-cell compartment, increased CD4+ and normal CD8+ T-cell numbers have been found, whereas both subpopulations were found to be decreased in another study. To date, however, studies on CD4+ T-cell subpopulations, T-cell proliferation history, and T-cell diversity are lacking [van deer Weerd et al. 2012; https://www.ncbi.nlm.nih.gov/pmc/articles/PMC3266399/]. A study conducted by van deer Weerd et al. [2012] with a total of 13 morbidly obese (BMI> 40 kg / m2) and 25 lean (BMI <25 kg / m2) subjects showed that T-cell counts were significantly increased in morbidly obese patients. CD4+ T cell counts increased while CD8+ T cell counts remained normal. This resulted in an increased CD4/CD8 ratio (morbidly obese 2.82 [1.62–6.17] vs. lean 1.54 [1.29–5.23]. In our previous research the frequency of the CD4/CD8 ratio was also contingent on the body fat content and in all our study participants (both active and inactive), high fat content shifted the CD4/CD8 ratio <1 [Tylutka et al. 2021]. So, changes in T cell subpopulation in obese individuals are characterized by increased homeostatic proliferation of both CD4+ and CD8+ T cells, possibly due to cytokines such as IL-7 and CCL5. This increased homeostatic proliferation is associated with an increase in CD4+ T cells. Changes in the CD4/CD8 ratio, in addition to fat content, can also be influenced by gender, age, ethnicity, genetics, exposure and infectivity, which can explain these discrepancies [McBride and Striker 2017; https://pubmed.ncbi.nlm.nih.gov/29095912/, Tylutka et al. 2021; https://pubmed.ncbi.nlm.nih.gov/33752623/].
Discussion
P12, L337, “Our research showed that the IgG CMV+ elderly 337 diagnosed with hypertension achieved higher values of CD4+ T cells compared to the con-338 trols, which may indicate that the increase in CD4+ T cells in hypertensive patients is in-339 dependent of CMV infection.”. Although it seems that the result includes confounding factors such as BMI, FMI, and sex, is there no bias or problem in your result? Will a statistical test like ANCOVA which can remove confounding factors show the same result as your one in this manuscript?
Thank you for pointing this out. According to the suggestion, the analysis of covariance (ANCOVA) was used in classification of body composition and gender that might influence the concentration of T lymphocyte phenotypes in individuals with diseases. We showed that the percentage of CD4+T lymphocytes in hypertensive patients was also observed to depend on anthropometric parameters (BMI, FM and FMI) and gender, and interestingly, men with hypertension showed a similar number of CD4+ lymphocytes to the percentage recorder in women who had not been diagnosed with hypertension. We added a description of the ANCOVA analysis in section 2.8 statistical analyzes, and the obtained results have been presented in Figure 3 and described in section 3.2 flow cytometry analyzes and in the 4. discussion section.
P12, L374, “The recorded differences between countries/regions can possibly be related to lifestyle factors such as habits, nutrition as well as physical and mental health status.”. Is there any evidence to support this hypothesis?
During the past decade, three prospective cohort studies with the participation of Swedes, Dutch and Belgians were performed to assess the IRP in the older adults defined by the CD4/CD8 ratio [Adriaensen et al. 2015; https://pubmed.ncbi.nlm.nih.gov/24568932/]. The inconsistencies may be ascribed to a large number of factors, including gender, age, nutrition, amount of physical activity or fat content, which can all affect the ratio, and also to the fact that the values ³1 or ≤2.5 are commonly used as the reference values in healthy individuals [McBride and Striker 2017]. The CD4/CD8 ratio can also be a useful marker to determine the body response to lifestyle exercise. Researchers have not yet unequivocally established whether the CD4/CD8 ratio increases or decreases with age. Neither are they unanimous as to whether the rise or the decline in the ratio is more favorable to maintain the longevity of the older adults. The CD4/CD8 ratio was found to increase with age in OCTO/NONA surviving participants over 100 years of age [29]. On the other hand, the analysis by Vasson et al. [2013] showed a decreasing trend of the CD4/CD8 with age in Spanish and French population. In our previous studies, we searched for the answer whether lifestyle exercise had an effect on the CD4/CD8 ratio. Interestingly, the CD4/CD8 ratio was found to fall within the range of the reference values in 55.9% of the group of older active participants. Our study group of active older adults was classified as representing healthy ageing. The frequency of the CD4/CD8 ratio is also contingent on the body fat content and in all our study participants (both active and inactive), high fat content shifted the CD4/CD8 ratio <1 [Tylutka et al. 2021].
The following sentence has been modified and now its reads as follows: „…Vasson et al. [10] also analyzed the biomarkers of immune status in 300 healthy volunteers aged between 20–75 years recruited in Austria, Spain and France and they compared three European countries and changes in T, B and NK cells. The immune parameters in French volunteers were recorded as constantly intermediate values in comparison withthose measured in Austrian and Spanish individuals, thus reflecting the importance of a geographic variability on the immune status and possible gradual differences in the dietary, lifestyle habits, environmental, genetic and socio-economic factors between Southern and Northern Europe. What’s more, they also noticed changes in CD4/CD8 ratio which differed significantly in Austria (1.8 ± 0.1), in France (1.5 ± 0.1) and in Spain (1.2 ± 0.1).In turn, in our Polish population we showed the statistically significant differences in CD4/CD8 ratio in the IgG CMV+ elderly aged 60-74 years in comparison to the IgG CMV+ elderly aged 75-90. Interestingly, our research findings contradicted the results reported by Adriaensen et al. [2015], in which a decrease in the CD4/CD8 ratio was observed in patients with IgG CMV+. The differences between countries/ regions are not only related to lifestyle factors such as eating habits, but also to lifestyle exercise and body fat content as shown in our previous research [Tylutka et al. 2021].”
Minor comments
- Materials and methods
2.1. Participants
P2, L78. Could you describe the criteria for hypertension?
The criteria for hypertension were based on medical records and the medical interview performed by the primary care physician who was engaged in the study. The suggested content has been added to the manuscript.
P2, L94. How did you define and calculate “the repeatability”? Moreover, how many people did participate the measurement?
Body composition measurements have been performed twice in each patient and the mean values have been added to the manuscript. 98% of the results of the second measurement have been identical to those in the first measurement.
2.2. Body composition
P3, L93. “7:00 and 9:00” is in the morning or afternoon? Why did you determine to measure at the time? Additionally, was the measurement performed before eating something?
Thank you very much for this remark. The description has been revised and it reads as follows: „Measurements were taken for each individual twice in a standing position between 7:00 and 9:00 am before the first meal. The participants were before blood sampling, after a night’s rest and with an empty bladder. The participants were instructed not to exercise vigorously for 12 hours prior to the analysis, as the time is needed for the purpose of recovery„
P3, L95. Which part in your manuscript did follow the methods of Tylutka et al.
Thank you for this question. The area of our scientific interests is related to both: immunosenescence and inflammaging. We used similar but not the same analytical methods in both articles analyzing the two subject matters. The similarity in the description of the research tools used results from the wide spectrum of research conducted with the participation of elderly individuals representing healthy or morbid aging. However, we would like to note that in the article entitled "Lifestyle exercise attenuates immunosenescence; flow cytometry analysis” published in BMC, the main aim of the article was to assess the impact of physical activity on the attenuation of immunosenescence. Eventually, the study included 54 elderly individuals who were classified into two groups: active vs. inactive based on the gait speed and the results of cardiorespiratory fitness. In the article "Pre-existing hypertension is related with disproportions in T-lymphocytes in older age." we did not address the issue of physical activity, whereas now in our study which included99 elderly participants (n=51 with hypertension and n=48 without hypertension control group), we discussed the impact of immune aging in terms of age, gender, and comorbidities, mainly hypertension diseases.
P3, L98. Is this criterion appropriate to the participants in this manuscript? For example, Asian people are relatively smaller than other people, so it seems to tend that the BMI of Asian people is underestimated. Does this effect include in your study?
This study was designed to evaluate the effect of common comorbidities in older age including hypertension on the changes in the T cell subpopulation and to determine the direction of changes in CD4/CD8 ratio and CMV infections in Polish population aged over 60 years. The inclusion criteria were as follows: an informed consent signed by all the participants, 60-90 years of age to be classified as an older group, their mobility, no hospitalization within6 months before the study and the same access to medical care. The study excluded elderly individuals with neurological disorders, acute infectious and oncologic diseases or with an implanted cardiac pacemaker (body composition analysis could not be performed). We absolutely agree with the reviewer that BMI values depend on nationality. Our research included only people from Poland, so the BMI values: (18.5 to 24.9 kg/m2)for the normal weight,(25 to 29.9 kg/m2) for overweight ( ≥ 30 kg/m2) and obesity, are in line with the WHO values for Europeans [Flegal et al. 2014; https://www.ncbi.nlm.nih.gov/pmc/articles/PMC4732880/].
2.4. Flow cytometry analysis
P4, Figure2. What does the color mean? Additionally, could you describe the meaning of purple values in a figure.
The results of the cytometric analyzes were processed using the Flowjo 9.6.6 cytometric analysis program, and the purple values shown in Figure 2 are the percentages of the individual cell populations.
2.5. Cytomegalovirus (CMV) IgG
P4, L136. How population was used and CV calculated?
The values of inter and intra assay CV are given in the determination methodology (ELISA, DRG, EIA3468). 12 measurements have been madein intra assay, and 3 measurements were taken in inter-assay; the mean values for inter and intra assay have been included in the paper. Here follows the print screens and a brief explanation of the calculations.
- Intra - assay
9,66+9.89+9.41+9.36+7.47+9.95+4.33+9.61+4.35+4.80+5.24+8.92=92.99/12=7.75%
2) Inter - assay
13.01+10.46+10.90=34.37/4 =11.45%
2.7. Biochemical variables
P4, L136. How population was used and CV calculated?
The values of inter and intra assay CV are given in the determination methodology: https://sceti.co.jp/images/psearch/pdf/DRG_EIA3954_p.pdf. In both cases, 5 determinations have been made, and in this article the mean values for inter and intra assay have been included. Here follows the print screens and a brief explanation of the calculations.
1) Intra - assay
7.5%+4.1%+4.2%+4.1%+2,3% = 22.2/5 =4.44%
- Inter -assay
4.1%+2.5%+4.1%+3.2%+2.5%=16.4/5 =3.28%
2.8. Statistical analysis
P5, L164, 165. “no effect if 0 ≤ η2< 0.01, a small effect if 0.01≤ η2<0.06, a moderate effect if 0.06 ≤ η2< 0.14, and a large effect if η2 ≥ 0.14.” Could you show the reference?
As suggested by the Reviewer, this section has been revised and it reads as follows:„ Additionally, eta-squared for ANOVA (η2) was used as a measure of the effect size which is indicated as having no effect if 0 ≤ η2< 0.01, a minimum effect if 0.01 ≤ η2< 0.06, a moderate effect if 0.06 ≤ η2< 0.14, and a strong effect if η2 ≥ 0.14 [Miles et al. 2001, Cohen et al.1988]. The effect size for Kruskal-Wallis test as the eta squared (η2) was calculated based on the function in Rstudio: kruskal_effsize and = 0.01-0.06 was a small effect, 0.061-0.14 was a moderate effect and > 0.14 was a large effect [Radovanović et al. 2020].”
P165, 167. Please italicize the symbols of spearman rank correlation coefficient rs and statistical significance p.
Thank you for pointing this out. The symbols of spearman rank correlation coefficient and p-value has been italicized consistently throughout the manuscript
- Results
3.1. Body composition
P5, L178, Table 1. Could you describe the number of males and females, BMI average, the number of BMI normal, overweight, obese, and the number of thyroid diseases, rheumatoid arthritis, diabetes for each group in the table.
The percentage of overweight, obese and normal BMI individuals for hypertensive patients and controls has been added in Table 1. The exact number of the elderly with the diseases has been given in Figure 1.
P5, L178, Please italicize the symbols of statistical significance p.
The symbol of p-value has been italicized consistently throughout the manuscript
3.2. Flow cytometry analysis
P6, L196, Table 2. Could you show the CD4/CD8 ratio within the reference range (≥1 or ≤2.5) and out of reference range (<1 or >2.5) for hypertension and control population in the table.
According to the Reviewer’s suggestion, Table 3 has been complemented with the percentage of CD4/CD8 within the reference range (≥1 or ≤2.5) and out of reference range (<1 or >2.5) for hypertension and control group.
P7, L226, Figure 3. Could you explain the boxplot in detail (e.g., the meaning of box, black line in the box, error bar).
Figure 3 (now Figure 4) shows the immune cell analyzes (%) i.e. CD4+, CD4+CD45RA+for the two groups: 60-74 years (yellow box) and 75-90 years (green box). The black line stands for the mean value and the long error bar is the standard deviation.
3.3. CMV IgG status and immune cells
P9, L262, Figure 5. Please italicize the symbols of statistical significance p.
The symbol of p-value has been italicized consistently throughout the manuscript
P10, L268, Figure 6. Please italicize the symbols of statistical significance p.
The symbol of p-value has been italicized consistently throughout the manuscript
3.4. Haematological variables
P10, L281, Table 3. Please italicize the symbols of statistical significance p.
The symbol of p-value has been italicized consistently throughout the manuscript
3.5. Biochemical variables
P11, L299, Table 4. Could you show the reference source for “Reference values”? Are the reference values not dependent on age and sex and suitable for your data?
Lipoproteins, i.e. total cholesterol, triglycerides, LDL and HDL, were performed using Biomaxima reagents dedicated to the BM200 apparatus - none of these reagents took into account the reference values by age or gender. CRP was determined using ELISA DRG in accordance with the methodology: https://sceti.co.jp/images/psearch/pdf/DRG_EIA3954_p.pdf also reference values for sex are not taken into account only for age: Neonatal serum: 0.01 to 0.35 mg / L; Adult serum: 0.068 to 8.2 mg / L. Glucose concentration was determined using Diaglobal test and there were no references value for sex and gender.
- Discussion
P11, L315, The reference [22-25] targeted participants around 50-60 years old. Therefore, although the participants in this manuscript are much older than those in previous studies, does the reference result apply to your result?
Thank you for pointing this out. Hypertension is a very common cardiovascular disease that has a significant impact on public health. It affects around 30% of patients in Europe and the United States. In Poland, according to the RYZYKO programme, 36.8% of patients suffer from the disease. The prevalence of hypertension increases with age, and it will pose an ever-increasing challenge for our ageing society [Reiwer-Gostomska et al. 2019;file:///Users/annatylutka/Downloads/64619-191080-1-PB.pdf]. Nevertheless, so far, there have not been many studies related to the influence of T lymphocytes on the development of arterial hypertension in the elderly. 99 elderly individuals took part in our study, including 51 people aged: 72.3±5.9 diagnosed with arterial hypertension. In our study, we observed an increase in the percentage of CD4+ T cells as well as CD4+ memory T cells in patients with hypertension compared to those without hypertension. Both Itani et al. [2016] and Ni et al. [2017] studies confirmed our outcomes, but we all absolutely agree with the Reviewer that we cannot compare these observations in terms of age as both Itani et al. [2016] and Ni et al. [2017] studies concern elderly people who were younger than our research group. Nevertheless, these studies can confirm that in the case of hypertension, T-lymphocytes play an important role, regardless of age, which should also be taken into account in future therapies. According to the Reviewer’s suggestion, this section of the discussion has been revised to clarify the age of the respondents.
P12, L327, “Thus, these findings suggest that CD4+ T cells and can serve as markers and biomarkers in the development of hypertension and might be therapeutic targets for this widespread disease.” Does this sentence apply to anyone of any age?
So far, most of the research conducted with regard to the effects of hypertension on the immune system has used animal models. In many experimental models of hypertension including genetic model and salt or angiotensin (Ang II)-induced model, the key role of T cells has been demonstrated [Mikolajczyk et al. 2019; https://www.ncbi.nlm.nih.gov/pmc/articles/PMC6647517/]. Itani et al. [2016;https://pubmed.ncbi.nlm.nih.gov/27217403/] have used a humanized mouse model in which the murine immune system was replaced by the human immune cells. They observed increased infiltration of human leukocytes, T cells, and especially CD4+ subsets in thoracic lymph nodes, thoracic aorta, and kidney in response to Ang II infusion. Also, CD8+ infiltration was higher in both lymph nodes and thoracic aorta in hypertensive animals compared with normotensive. The increase in memory T cells CD3+CD45RO+ was noted in the aortas and lymph nodes. In this model, human T cells become activated and invade end-organ tissue, in response to Angiotensin II stimuli. They also analyzed changes in T cells in 20 hypertensive patients aged: 52.6 ± 11 then compared to control (normotensive) patients aged: 52.6 ± 12. Hypertensive patients had significantly higher office blood pressures and both systolic and diastolic blood pressures as measured by ambulatory blood pressure monitoring. Plasma renin and aldosterone levels were elevated in the hypertensive group compared to the normotensive subjects. It was shown that the percent of both CD4+ and CD8+/CD45RO+ circulating T cells was greater in the hypertensive patients than in the normotensive controls.
In response to the Reviewer’s question, the following sentence has been rewritten: „…Youn et al. [22] showed a significantly higher number of circulating immunosenescent pro-inflammatory CD8+ T lymphocytes in n= 71 individuals with hypertension aged 51.6±11.2 compared to n=71 healthy ones aged 51.5±12.2. Interestingly, in our study, we observed higher numbers of the CD4+ T lymphocyte in individuals with hypertension compared to the controls. Ni et al. [23] analyzed 40 individuals with essential hypertension (EHs) aged 56.14±2.19 years and 40 normotensive healthy participants (NTs) aged 53.60±3.45 years and also observed a higher percentage of CD3+CD4+ in EHs than in NTs patients. In the available literature flow cytometry analyses have already revealed an increased infiltration of leukocytes (CD45+) and CD4+ lymphocytes in response to the infusion of angiotensin. We also observed an increasing CD4+ memory T lymphocytes in hypertensive participants. Itani et al. [24] showed higher values of circulating CD4+ and CD8+ memory T lymphocytes in hypertensive patients aged: 52.6 ± 11 compared to normotensive control aged: 52.6 ± 12 Research results may suggest an important role of T lymphocytes in the development of hypertension in different age groups and could be used as therapeutic targets in this widespread disease in the future.”.
P12, L330-332, “several age-related changes in immune functions can be linked to longevity and the predictors of mortality include increased IL6 levels and the ratio CD4/CD8 <1.”. As Figure 3 shows, Although CD4/CD8 ratio decreases with aging, did your result match the above previous result?
Thank you for pointing this out. In our study, we analyzed changes in the CD4/CD8 ratio depending on gender, age and comorbidities. As mentioned above, many factors such as gender, age, and exposure to pathogens can influence the CD4/CD8 ratio, and both increased and decreased ratios can be related to the immune risk phenotype. Alonso-Fernandez & De la Fuenta et al. [2011] observed that one of the predictors of mortality is CD4/CD8 <1 and an increase in IL-6. Interestingly, in our study, we observed an increase in CD4/CD8 with age (Figure 4). It is also interesting that patients with hypertension also demonstrated an increase in CD4/CD8 and only 4% have a CD4 / CD8 ratio <1.The following sentence has been revised and it reads as follows: „According to Alonso-Fernandez & De la Fuenta [25], several age-related changes in immune functions can be linked to longevity and the predictors of mortality include increased IL-6 levels and the ratio CD4/CD8 <1. In our study, CD4/CD8 ratio >2.5 was observed in approx. 34% of the participants with hypertension, and only 4% had CD4/CD8 <1. Ni et al [23] also noted a higher CD4/CD8 ratio as well as serum IFN-γ and TNF-ɑ levels in essential hypertension compared to normotensive individuals”.
P12, L333-334, “the same observations in patients with essential hypertension compared to normotensive individuals were reported by Ni et al. [23].” How proportion of CD4/CD8 ratio >2.5 previous study observed. As I see the reference paper written by Ni et al. [23], although it looks maximum CD4/CD8 ratio is around 2.5, did your result in patients with hypertension, “CD4/CD8 ratio >2.5 was observed in approx. 34% of the participants with hypertension”, show the same observations with the previous result?
Thank you for pointing this out. The following sentence has been changed into: „In our study, CD4/CD8 ratio >2.5 was observed in approx. 34% of the participants with hypertension, and only 4% had CD4/CD8 <1. Ni et al. [23] also noted a higher CD4/CD8 ratio as well as serum IFN-γ and TNF-ɑ levels in essential hypertension compared to normotensive individuals.”
P13, L398, “small sample size”. Please describe the sample size for each gender to analyze the gender-related trends of changes in T lymphocytes population.
The main focus of our study was on the influence of arterial hypertension on changes in the T lymphocyte subpopulations .Additionally, we also wanted to observe changes in the studied subpopulations depending on gender and age. Our study group consisted of n= 99 elderly individuals, n = 51 of whom had arterial hypertension and n = 48 constituted the control group (without hypertension). The groups were similar in size. When calculating the sample size for the whole group, assuming that 950 elderly people attend the University of the Third Age (U3A), assuming a confidence interval of 10%, the minimum number of respondents should be 87 people. Out of the 99 people who took part in the study, n= 83 people were women and n = 16 were men. Taking into account the total number of men attending U3A n = 200 and the confidence interval of 10%, the number of men to draw reliable conclusions should be at least n=65 individuals. As we pointed out in the discussion, on the basis of the observed changes in the subpopulation of T lymphocytes, it is not possible to draw unequivocal conclusions, due to the fact that the male sex group is too small. Consequently, the limitations section has been rewritten to read as follows: "The limitations of the study include a relatively small number of participants, especially unequal proportion of gender, and no information on their lifestyle and environmental factors"

Round 2
Reviewer 2 Report
This manuscript has been significantly revised and improvements in the content have been noted. Therefore, the reviewer judged it was suitable for acceptance.